# The High-Elevation Peatlands of the Northern Andes, Colombia

**DOI:** 10.3390/plants12040955

**Published:** 2023-02-20

**Authors:** Juan C. Benavides, Dale H. Vitt, David J. Cooper

**Affiliations:** 1Departamento de Ecología y Territorio, Pontificia Universidad Javeriana, Bogotá 110231, Colombia; 2School of Biological Sciences, Plant Biology, Southern Illinois University, Carbondale, IL 62901-6509, USA; 3Department of Forest and Rangeland Stewardship, Colorado State University, Fort Collins, CO 80523-1572, USA

**Keywords:** Andes, *Distichia muscoides*, human disturbances, mountain peatlands, peatlands, plant diversity, *Sphagnum*

## Abstract

Andean peatlands are important carbon reservoirs for countries in the northern Andes and have a unique diversity. Peatland plant diversity is generally related to hydrology and water chemistry, and the response of the vegetation in tropical high-elevation peatlands to changes in elevation, climate, and disturbance is poorly understood. Here, we address the questions of what the main vegetation types of peat-forming vegetation in the northern Andes are, and how the different vegetation types are related to water chemistry and pH. We measured plant diversity in 121 peatlands. We identified a total of 264 species, including 124 bryophytes and 140 vascular plants. We differentiated five main vegetation types: cushion plants, *Sphagnum*, true mosses, sedges, and grasses. Cushion-dominated peatlands are restricted to elevations above 4000 m. Variation in peatland vegetation is mostly driven be elevation and water chemistry. Encroachment of sedges and *Sphagnum sancto-josephense* in disturbed sites was associated with a reduction in soil carbon. We conclude that peatland variation is driven first by elevation and climate followed by water chemistry and human disturbances. Sites with higher human disturbances had lower carbon content. Peat-forming vegetation in the northern Andes was unique to each site bringing challenges on how to better conserve them and the ecosystem services they offer.

## 1. Introduction

Peatland plant communities are strongly controlled by hydrological regime, flora, water chemistry, site history, and climate [1,2,3]. In the northern tropical Andes, mountain peatlands are common features of the landscape. Peat-forming plants colonize and grow in mountain areas with high precipitation, low evapotranspiration, and complex landforms that allow for the accumulation of water in natural depressions [4,5]. In the páramos, tropical alpine areas in the northern Andes, plant communities vary with elevation, highlighting the relevance of the temperature–radiation gradient on their distribution [6,7]. The tropical Andes are a biodiversity hotspot supporting nearly 15% of global plant species. An important source of the high number of species is the diversity of habitats and microhabitats, as well as the recent evolution of plant species as the Andes developed [8,9]. In the Andes, peatlands provide distinctive habitats for important species, many of them endemic [1,10,11,12,13]. 

The tropical Andes have been intensively used for cattle grazing and agriculture for the last 200 years, and only the regions at the highest elevations have remained isolated from human disturbances, largely due to their harsh conditions. However, rising temperatures are expected to allow for human development in areas that were previously unproductive [14,15]. The expansion of human disturbances will introduce changes in the natural páramos vegetation affecting key ecological processes, such as hydrological cycles and nutrient and carbon storage [16,17,18]. Additionally, peatlands are important in climate change mitigation. Globally, peatlands are a major carbon stock, storing approximately one-third of the world’s soil carbon (450 Pg) in only 3% of the world’s continental area [19,20]. However, climate change and the changes in land use observed during the last 20 years have increased the vulnerability of peatlands to disturbances [21,22,23,24].

In general, peatlands form in wetland habitats where plant production exceeds decomposition due to waterlogging over long time periods, allowing for the accumulation of an organic soil [25,26]. Peat is produced by plant litter—stems and roots that have a slow decomposition, either by the particular properties of the dead plant tissues, or by the anoxic and acidic conditions of the saturated soils [27]. In peatlands, only a fraction of the plants on the surface have the ability to form peat, and these usually become the dominant species [28,29]. Andean peatlands have unique vegetation types that are able to accumulate peat in small depressions or on large plateaus with limited drainage under strong climatic and hydrological gradients varying from regional climates—i.e., wet in northern regions and arid in the central and southern Andes—and from the upper tree line to extreme high elevations [1,10,12]. Peatlands are dominated by cushion plants in the central and southern Andes, with sporadic occurrences of vegetation types dominated by mosses or sedges in the northern and southern ends of the range [1,30]. Although the influence of large climatic variations has been addressed in the past, those studies have been focused on cushion-dominated peatlands in the central and southern portions of the Andes [12]. In the northern Andes, a diversity of vegetation types has been observed, with influence from fluctuations in the water table and water chemistry. Peatlands with an abundance of *Sphagnum* are found in more acidic conditions, with sedges dominating in sites with higher pH [1]. However, distinctive peatland systems can be found in extreme habitats at very high elevations, i.e., regions with freezing temperatures every night [31]. Preliminary classifications of peatland vegetation in the northern Andes have identified four main vegetation types: cushion-plant-dominated, sedge-dominated, *Sphagnum*-dominated, and tussock-grass-dominated peatlands [1,6]. Differentiation among the Andean peatland types has been proposed to come from variation in elevation and water chemistry. *Sphagnum*-dominated peatlands are usually more common in acidic and nutrient-poor habitats, sedges and tussock grasses are common in nutrient-rich habitats with fluctuations in the water table, and true mosses are dominant in sites with high nutrient concentration, slightly acidic pH, and low variation in the water table [32,33]. Peatlands dominated by cushion plants are found only at very high elevations in the northern and central Andes above 4.000 m in elevation [1,10,34].

Our understanding of how the main vegetation types of peat-forming vegetation form along environmental gradients is limited mainly to high-elevation cushion-dominated peatlands in the central and southern regions of the Andes. Higher variation of peat-forming vegetation types has been observed in the wetter regions of the northern Andes. Information about the effect of human disturbances on different peatland types in the northern Andes is lacking. The present research is focused on understanding how northern Andean peatland vegetation types vary along the combined gradients of water chemistry and climate. We addressed two research questions: (1) How does the variation in the main peat-forming vegetation types relate to elevation and water chemistry in the northern Andes; (2) How do human disturbances affect plant composition in tropical Andean peatlands and what are the possible implications for peat functioning. 

## 2. Results

### 2.1. Plant Diversity and Vegetation Types 

We found 264 plant species at the 121 peatlands surveyed; 124 species were bryophytes, 140 were vascular plants, and 77 (29%) of the species were found only at one site. There was an average of 18 species (±6) at each peatland (10 vascular plants and 8 bryophytes). When vascular plants and bryophytes are considered together, sites having more species occur at elevations between 3500 and 4000 m (Figure 1) (Species = a + b * elevation + c * elevation^2^, R^2^ = 0.12, *p* < 0.001). Relevant peat-forming species were species of *Sphagnum*, with a predominance of *Sphagnum sancto-josephense* at sites with evidence of disturbance; the sedge *Carex pichinchensis*, common on minerotrophic sites; and the cushion plants *Distichia muscoides* and *Plantago rigida*, more common at higher elevations. Species of particular relevance were the mosses *Sphagnum antioquensis* and *S. cundinamarcanum*, previously known only from the original collections, and a new species of *Colura*, the latter a liverwort usually found as an epiphyll but here found growing on cushions of *D. muscoides* at 4300 m.a.s.l. [35,36]. *Carex bonplandii* was the most common species in sedge- and true-moss-dominated peatlands. Other species found were the bryophytes *Hypnum cupressiforme* and *Bryum laevigatum*, and the vascular plants *Hypericum juniperinum* and *Diplostephium vaccinoioides*. Several species were shared between sedge- and *Sphagnum*-dominated sites, including *Sphagnum sancto-josephense* and the vascular plants *Cortaderia nitida*, *Carex tristicha*, *Arcytophyllum muticum*, and *Blechnum loxense*. 

Our preliminary k-means cluster analysis indicated that classifying the sites into five groups provided the highest SSI. The five clusters corresponded with the dominance of different peat-forming plant species and had significant differences in their dispersion (PERMANOVA test F_3,117_ = 11.2, *p* < 0.001, Appendix A). The five clusters were named according to the dominant vegetation observed: (1) *Sphagnum*-dominated sites, (2) true mosses (Bryopsida), (3) grasses, (4) sedges, and (5) cushion plants (Figure 2). The number of plots (*n*) and total number of species or gamma diversity (S) for the different vegetation types are provided.

The *Sphagnum* cluster (*n* = 26, S = 109) was dominated by several *Sphagnum* species, including *S. sancto josephense*, *S. falcatulum*, *S. magellanicum*, and *S. oxyphyllum* together with the grass *Cortaderia nitida*. *Juncus effussus* and *S. sancto-josephense* were the most common species in this vegetation type occurring in 19 and 16 of 26 sites. Indicator species for the Sphagnum cluster were the vascular plants *Juncus effusus*, *Paepalanthus colombianus*, and *Nertera granadensis*. *Sphagnum* species were common, and their indicator value was low. The *Sphagnum*-dominated sites had large variation on the shrub and herbaceous layers ranging from sites having lawns of *Sphagnum* with almost no herbaceous vegetation to sites with a complete shrub cover of *Diplostephium* (Asteraceae) or *Escallonia* (Escalloniaceae) (Table 1). 

The true moss cluster (*n* = 31, S = 158) was dominated by *Breutelia chrysea*, *Pleurozium schreberi*, and *S. sancto-josephense.* Indicator species for the true moss vegetation type were the herbaceous vascular plant *Arcytophyllum muticum* (Rubiaceae) and the mosses *Pleurozium schreberi*, *Breutelia chrysea*, and *Hypnum cupressiforme*. The structure of the vegetation was more uniform than the *Sphagnum* sites, with a ground layer dominated by mosses and an herbaceous layer with sedges and grasses but lacking a developed shrub layer. The microtopography was complex with large hummocks made by a combination of small shrubs and *P. schreberi* and hollows with *B. chrysea*, with a lack of standing water in the hollows (Table 1) (Figure 2). 

The sedge cluster (*n* = 22, S = 93) was dominated by the sedge *Carex bonplandii*, the mosses *Sphagnum sancto-josephense* and *S. magellanicum*, and the ericaceous shrubs *Pernettya prostrata* and *Disterigma empetrifolium*. Indicator species for the sedge cluster were the shrub *Pernettya prostrata* and the rosette species *Puya goudutiana* and *Espeletia grandiflora*. The structure of the vegetation of the sedge vegetation type was dominated by extensive lawns of sedges that grow in tussocks, leaving small and wet hollows *with S. magellanicum* in the hummocks and *S. sancto-josephense* in the hollows (Table 1) (Figure 2). 

The grasses cluster (*n* = 18, S = 84) was dominated by the small sedge *Carex pygmaea*, followed by larger herbaceous species such as *Calamagrostis effussa* and *Rhynchospora oreoboloidea* and the ericaceous shrub *Pernettya prostrata*. *Sphagnum* cover was low. Indicator species for the grass cluster were the sedges *Carex pygmaea*, *Oerobolus*, and *Rhynchospora oreoboloidea*. Where *Oreobolus* species were dominant they were the main peat-forming plant species. 

The cushion plant cluster (*n* = 24, S = 102) was clearly identified in the cluster analysis and the NMDS ordination (Figure 2), the cushion-forming plants *Distichia muscoides*, *Plantago rigida*, and *Werneria pygmaea* being dominant. The cushions grow forming clear hummocks, and in the space between the cushion columns a number of bryophyte species occur, including the liverworts *Isotachis multiceps* and *Anastrophyllum nigrescens* and the moss *Warnstorfia exannulata*. Indicator species for the cushion plant vegetation type were the most abundant species, *Distichia muscoides*, *Plantago rigida*, and *Werneria pygmaea*. The cushion plants allowed for the growth of rare liverwort species among the tight spaces between their leaves with several rare species such as *Colura* sp nov., *Fossombronia* (*Austrofossombronia*) *peruviana*, *Riccardia regina*, and *Lobatiriccardia* sp. (Table 1) (Figure 3). 

### 2.2. Water Chemistry

Surface water pH varied from 3.5 to 7.5. Most sites (94) had a pH below 5.5 and *Sphagnum*-dominated sites were almost always found where the pH was below 5.5. Sites dominated by sedges had a wide variation in pH ranging from 3.5 to 7.5; acidic sites dominated by sedges were in sites with high precipitation facing the Pacific Ocean in the westernmost mountain range. Sites dominated by true mosses and cushion plants were less acidic; sedge- and *Sphagnum*-dominated sites had similar acidity (Figure 4, Appendix A). Electrical conductivity of the water was higher at cushion-dominated peatlands than the other vegetation types (Kruskal–Wallis chi-squared = 39.909, df = 3, *p*-value = 1.114 × 10^−8^) (Figure 4). Cushion plants preferred sites with water electrical conductivity from 80 μS cm^−1^ to 700 μS cm^−1^ in sites influenced by recent volcanic activity. Cation concentrations in pore water were higher in cushion-dominated peatlands than sedge-dominated or *Sphagnum*-dominated peatlands (Figure 4). Ca^++^, K^+^, Mg^++^, and Na^+^ concentrations were higher in cushion plants, with the lowest values in *Sphagnum*-dominated sites and sedges (Ca^++^ F_4116_ = 3.96, *p* = 0.004, K^+^ F_4116_ = 7.14, *p* < 0.001, Mg^++^ KW chi-squared = 54.14, df = 4, *p*-value < 0.001, and Na^++^ F_4116_ = 6.5, *p* < 0.001). The main trend observed was in elevation, with cushion plants at higher elevation followed by sedge-, true-moss- and *Sphagnum*-dominated sites at lower elevations (Figure 4). 

### 2.3. Peat Soil 

Peat bulk density in the 0–10 cm layer was higher for cushion plants and grass-dominated vegetation, although overall values were similar among the different vegetation types (KW = 26.24, *p* < 0.001) (Figure 5, Appendix A). Bulk density in the 10–20 cm layer and carbon content at the 0–10 and 10–20 cm intervals were similar among the different vegetation types. Carbon content was largely related to bulk density and similar for all vegetation types, except true-moss-dominated sites which had lower carbon content than the other sites (F_4115_ = 3.78, *p* = 0.006) reaching values of up to 40 tons per hectare in a 10 cm layer (Figure 5). Carbon storage in the 0–20 cm depth layer was larger in cushion- and grass-dominated sites with lower values in *Sphagnum*-dominated sites (F_4115_ = 9.04, *p* < 0.001). 

The mineral concentration in peat was higher in cushion plants than in the other vegetation types for K, Mg, and Na (K F_4108_ = 7.2, *p* < 0.001, F_4111_ = 3.7, *p* = 0.007, and F_4110_ = 4.74, *p* = 0.001) (Figure 6). Ca concentrations were higher than Mg, Na, and K in peat and were similar among the different vegetation types (Figure 6). Cation concentrations in water were related to cation contents in peat for K^+^ (r = 0.21, *p* = 0.01) and Na^±^ (r = 0.56, *p* < 0.001). The correlation between concentration of Ca^2+^ in water vs. peat was weak and non-significant (r = −0.008, *p* = 0.92). Overall, peat soils stored between 80 and 90% of their weight in water with similar capacity in the soils of the different vegetation types.

Cushion-dominated sites had an overall lower impact from human disturbance than the other sites (F_4110_ = 21.6, *p* < 0.001). An additional effect was observed between the intensity of the disturbance and the carbon content for all four vegetation types (F_1112_ = 20.16, *p* < 0.001) (Figure 7).

## 3. Discussion

Peatland vegetation is highly dependent on water chemistry, with a clear gradient of *Sphagnum* domination in mineral-poor sites to grass and sedge domination in more mineral-rich habitats [1,3,10]. In the Andes, peatland vegetation is highly variable due to complex interactions of the incoming water with different types of bedrock, nutrient runoff from hillsides, and climate [1]. Elevation plays an important role, acting as a key factor controlling the distribution of cushion plants with a clear limit at around 4100 m.a.s.l. [16]. Although hydrology is the main driver in peatland functioning, there is an internal feedback from vegetation that stabilizes the functioning of the system [38]. Peatlands are generally classified according to the mineral content of the water and water table stability. Rich fens in boreal areas are usually dominated by true mosses and sedges; however, minerotrophic *Sphagnum* species also form large colonies [39]. 

Our results show distinctive groupings of peat-forming species with a strong spatial component in five main vegetation types: cushion-plant-dominated, *Sphagnum*-dominated, true-moss-dominated, sedge-dominated, and grass-dominated. Generally, sites that are near each other will be placed under the same vegetation type but several sites distant from each other can be found in the same group. The spatial structure of the clustering indicates that large scale variables such as climate, geology, and disturbance regimes play a role almost as strong as the species’ niches in determining the peat-forming vegetation of a particular site [16]. The species *Sphagnum sancto-josephense* was common and found in all the vegetation types except sedges and cushions and in sites with very low to high pH and water electrical conductivity [16]. *Sphagnum*-dominated sites had a high diversity of *Sphagnum* species including endemics such as *S. cundinamarcanum*. Sedge-dominated sites had high diversity of sedge species producing a variable vegetation type, dominated by the species *Carex*, *Oreobolus*, and *Rhynchospora* in Cyperaceae and *Cortaderia* and *Calamagrostis* in Poaceae. These species have a broad distribution on the ordination reflecting the sharing of species between the sedge- and grass-dominated groups (Figure 2). A large number of species were found at only one site, with individual sites having a unique flora of rare and low-cover species where a combination of mountaintop isolation and unique climatic conditions in the inter-Andean valleys allows the development of distinctive vegetation types [1,6]. Overall conservation planning should take this high site uniqueness into consideration. The number of species found in the peatlands of the northern Andes is more than double the number found along a 30° latitudinal gradient in the southern Andes [10,12]. The number of species in peatlands from a small area in northern Peru was also lower, with 171 plant species compared to 265 in our study. Bryophyte diversity was also high, comparing the 124 found in this study with 69 found in northern Peru [1]. The greater difference in bryophyte species can, in part, be explained by the small size and taxonomic complexity of many liverworts [40].

*Vegetation*: Five main vegetation types identified in the cluster analysis were verified by the PERMANOVA (Appendix A). The lack of separation on the two-dimensional solution of the ordination highlights the sharing of common species such as *Sphagnum sancto-josephense*, *Carex bonplandii*, and *Cortaderia nitida* among sites that are not dominated by cushion plants. The five vegetation types responded to gradients of water chemistry, soil mineral content, and elevation. Cushion-plant-dominated sites had significantly different water chemistry compared to the other four vegetation types. In sites dominated by true mosses, human disturbance and pH were higher than in sites dominated by different species of *Sphagnum* (Figure 7). The presence of *Sphagnum sancto-josephense* in sedge, true moss, and *Sphagnum* vegetation types might be explained by the fact that *S. sancto-josephense* is related to minerotrophic species that tolerate high variation in pH and water table (Table 1) [38,41].

*Peat and water chemistry*: Higher water pH and electrical conductivity in cushion peatlands are possibly the result of abiotic factors such as geology or groundwater contact with the bedrock. Peatlands in the southern Andes with similar vegetation dominated by *Distichia muscoides* have pH and conductivity values in the same range as *Sphagnum*-dominated peatlands; pH reported from Patagonia and “Tierra del Fuego” peatlands varies from 3.8 to 4.5 [34,42]. Andean true-moss-dominated fens structurally resemble boreal rich fens; however, differing dominant species, much lower pH, and electrical conductivities provide a set of conditions making these Andean fens distinct from boreal rich fens. Our results highlight the lack of agreement between the vegetation and water chemistry approach used to classify the different types of boreal peatlands and Andean peatlands, since northern Andean peatlands’ ombrotrophic rain-fed bogs are lacking and there is no discernible effect of the water chemistry on the different vegetation types except for cushion-dominated sites at higher elevations.

*Peat carbon and water storage*: Carbon stocks in the upper 10 cm of soil were lower in *Sphagnum*-dominated areas than the other vegetation types and larger in cushion-dominated peatlands. Increments in bulk density have a multiplier effect on carbon stocks; thus, any increase in bulk density greatly affects carbon content estimates [27]. Organic matter and bulk densities of northern Andean peatlands are below the values reported for boreal peatlands. Mean bulk density for Canadian peatlands has been estimated at 0.09 g C cm^−3^ [43], similar to the values found in true-moss-dominated peatlands in this study (0.04 g C cm^−3^).

Functional aspects of peatlands are negatively affected by disturbance. Peatland drainage as a result of peat harvest, water redirection for agricultural irrigation, or human consumption affect peatland vegetation by allowing the encroachment of upland species that rapidly take over the previously saturated ecosystem [19,44]. In this study, highly disturbed peatlands had lower carbon content in the upper 10 cm of soil. In particular, mountain peatlands, despite their small size, are used as reservoirs for local water supply, either for human consumption in rural communities or agricultural irrigation with negative impacts on plant diversity and composition and recent peat accumulation rates [21].

In summary, the different vegetation types of high-elevation peatlands in the northern Andes are influenced by a combination of external processes, such as geology, climate, and regional disturbance regimes, and endogenous autogenic processes determined by the environmental requirements of the main peat-forming species and the ability of these species to form peat [16]. The sedge and *Sphagnum*-dominated sites store less carbon and are expected to have lower rates of carbon accumulation because of the lower bulk densities of bryophytes [45]. The elevation gradient has a sharp boundary at high elevations where only cushion plants are found with a clear dominance of *Distichia muscoides*. Cushion-dominated peatlands are common in the central and southern Andes and are mostly found in watersheds at high-elevation sites with exposed bedrock and low vegetation cover [1,46]. Peatland ecosystems in the northern Andes support unique biodiversity and offer important ecosystem services. However, human disturbances have a negative effect on carbon content in at least the upper 10 cm of soil, perhaps by increasing soil decomposition rates. Peatlands are either excluded or partially differentiated in the national wetland maps of countries in tropical America, and elements for their delineation such as vegetation, hydroperiod, and organic soil development are lacking [47,48]. More importantly, the unique biodiversity and species composition of each site creates challenges in finding how to better protect these ecosystems and the ecosystem services they offer.

## 4. Materials and Methods

### 4.1. Study Area and Site Selection

The Andes north of three degrees north latitude have mean annual precipitation of approximately 900 mm, varying from 600 mm y^−1^ to 3000 mm y^−1^ [6,49]. The cool temperatures and positive water balance provides conditions conducive to the development of peatlands, an ecosystem type that is common in the Andes and has been referred as “pantanales”, “pantanos” or “bofedales” by local communities [4,50]. High-elevation peatlands in the Andes initiated after the last glacial maximum around 17,000 ybp, most probably because of the wet and cool conditions that developed, and topographic depressions left after glacier erosion favored the development of peatlands [51,52].

Sampled peatlands were selected following a stratified sampling, choosing peatlands with contrasting characteristics of elevation, precipitation, and disturbance. Overall, 121 peatlands were sampled, covering a quadrangle of 400 km on each side that included the three main “Cordilleras” and most of the variation of the three major gradients explored: elevation, precipitation, and human disturbance (Figure 8). The sites visited were covered under research permit number 09_020409 from the Ministerio del Medio Ambiente in Colombia, 2009. The visited sites were selected following a hierarchical approach. First, we divided the regions from sites with low-to-high elevation since there was no reliable climatological data at the highest elevations. Geology was also a factor and sites from the three main Colombian cordilleras with contrasting geology were included. An additional filter was the research permit that authorized a subset of the sites. Finally, peatlands at each site were selected by having contrasting disturbance regimes and, if possible, contrasting vegetation types. Most of the sites were fairly remote, and it took between 1–2 days of hiking to reach each site.

The disturbance at each site was summarized in a single index combining the observed variables in the field: distance to nearest town, distance to nearest house, size of the town, number of adult people in the house, cattle and agriculture presence, abundance, and distance to nearest fields, presence and abundance of exotic or cultivated plants, and evidence of runoff from fields. We summarized all the variables in a single index using an inverted scale for the distances, assuming that the further away the town, house, or field, the less impact will occur to the ecosystem.

### 4.2. Field Data Collection and Laboratory Analysis

Plant diversity was surveyed using nested plots in each peatland, and at least one plot was located near the center of the peatland following the theory of centrifugal organization of wetlands [53]. Within each plot, tree-like and shrub vegetation canopy cover was visually estimated in 10 × 10 m plots, herbaceous cover in 2 × 2 m subplots located inside the 10 × 10 m plot, and bryophyte and ground layer cover in four 1 × 1 m subplots nested inside the 10 × 10 m plot. The 1 × 1 m subplots were located to cover a single microtopographic feature—either hummock or hollow—near the corners of the 10 × 10 m plot. Cover of all species was visually estimated to the nearest 5% and cover values estimated from the 1 × 1 m subplots were averaged and pooled and a single cover value was obtained for each species in each plot. Plant species when possible were identified in the field, with unidentified plants taken to the laboratory or sent to specialists to be identified. Plant vouchers are available at the SIU herbarium in Carbondale, Illinois, and Colombia National Herbarium (COL) in Bogota. Elevation was recorded for each site using a handheld GPS unit (60CSx Garmin). All bryophytes were collected and identified in the laboratory; vascular plants were identified by comparing with herbarium collections when the plant was not identified in the field. Regardless of our ability to identify each specimen to the species level, all the registered plants in this study were assigned to an operational taxonomic unit. The taxonomical units were differentiated using the main characteristics that separate species in each group from the specific literature for each group with emphasis on mosses, liverworts, and the vascular plants in the Asteraceae and Poaceae. Undetermined species were named following their corresponding family or genera followed by a number and all the specimens that were morphologically similar were assigned the same name (Appendix A).

Peat soil was sampled from 0 to 10 cm and 10 to 20 cm depths using a stainless-steel cylinder 10 cm long and 5 cm in diameter. The two peat samples were weighed in the field to the nearest gram and then oven dried in the laboratory at 60 °C for 48 h or until no change in sample weight was observed. The difference between wet and dry weight was the % water content. Organic matter content was estimated from dried samples using the loss on ignition method by combusting the peat samples at 400 °C for 16 h and reweighing [54]. The relationship between organic matter and organic carbon was estimated following known relationships [29]. Residual ash was solubilized using a strong acid digestion and taken to a known volume to estimate the main element concentrations of the peat [55]. The solubilized peat was analyzed for the major cations calcium (Ca^2+^), potassium (K^+^), magnesium (Mg^2+^), and sodium (Na^+^) using a Varian Spectra 220 atomic absorption spectrometer. Pore water was collected from fresh soil samples cored with a stainless-steel cylinder 10 cm deep and 5 cm in diameter. The sample was kept above the soil until excess water was drained without further mechanical extraction. Finally, using a manual vacuum pump the pore water was extracted from the soil sample in the field and filtered using a membrane filter with a pore size of 5 μm (Sigma-Aldrich millipore).

Water electrical conductivity and pH were measured at the center of each plot in two different open hand-dug pits that intersected the water table. Water electrical conductivity was corrected for the concentration of H^+^ [37] using a Thermo Scientific Orion 3-Star pH and conductivity meter. Samples of 125 mL of water were filtered with 4 µm pore diameter membranes within 24 h after sampling events. The water samples were taken to the laboratory and analyzed for major cation concentrations using the same method described above for peat chemistry. Water table depth relative to the surface from two pits and hummock–hollow variation from 20 different points in the 10 × 10 plot were recorded for each plot. Water storage capacity was obtained from the peat soil cores by subtracting the dry weight of the sample from the weight of the sample saturated with water, divided by the sample dry weight.

### 4.3. Analysis

The main vegetation groups were identified in an analysis using a k-means cluster analysis (tell us what program you used). Data were analyzed using the statistical software R version 4.02 and by customizing available statistical and graphical functions [56,57,58]. We selected the best number of groups by using the classification with the largest Simple Structure Index [58]. Each peatland and plot were assigned to a specific group of the k-means clustering followed by an indicator species analysis of each group [58] (Figure 2). The groups identified by the cluster analysis were tested for significant differences in their centroids and dispersion using PERMANOVA analysis due to its robustness against differences in dispersion within the groups [59]. All cluster and ordination analyses were done after removing the species that occurred only in one site.

The response of the different vegetation types and plant composition in each peatland to the environmental gradients was analyzed using non-metric multidimensional scaling (NMDS) based on the Bray–Curtis ecological distance matrix, and using the percent cover of each species in each plot as the response variable. NMDS is an unconstrained ordination method that preserves to the best possible extent the rank of the similarities between samples, in this case Bray–Curtis dissimilarity index, and represents such similarities in a predefined *n*-dimensional space [60]. Environmental variables were added to the ordinations using the vector fitting technique with 1000 permutations. Only vectors with *p* values below 0.01 were retained for further analysis; the angle of the vector indicates the direction of change of the variable and the length of the vector represents the degree of correlation of the projected variable with the ordination axis [58,61].

Differences in peat carbon content, peat and water chemistry, and climate were analyzed among the different vegetation types using one-way ANOVA, with post hoc Tukey’s HSD test to identify homogenous groups. ANOVA assumptions were checked for all variables: Normality of the residuals from the ANOVA analyses was tested by examining the normal scores versus ordered residuals plot or QQ plot, and homogeneity of variances was checked using the Levene test [62]. Cation concentrations in water were correlated with cation contents in peat using Pearson correlation. The Pearson correlation coefficient is assumed to have a t distribution, and a confidence interval is calculated [62]. Data were analyzed using the statistical software R version 4.02 and by customizing available statistical and graphical functions [56,57,58]. Bryophyte and vascular plant nomenclature follows the international plant name index (https://www.ipni.org/, accessed on 15 December 2022) and the Tropicos database (https://www.tropicos.org/, accessed on 15 December 2022) for taxonomically rare and new species [35].

## Figures and Tables

**Figure 1 plants-12-00955-f001:**
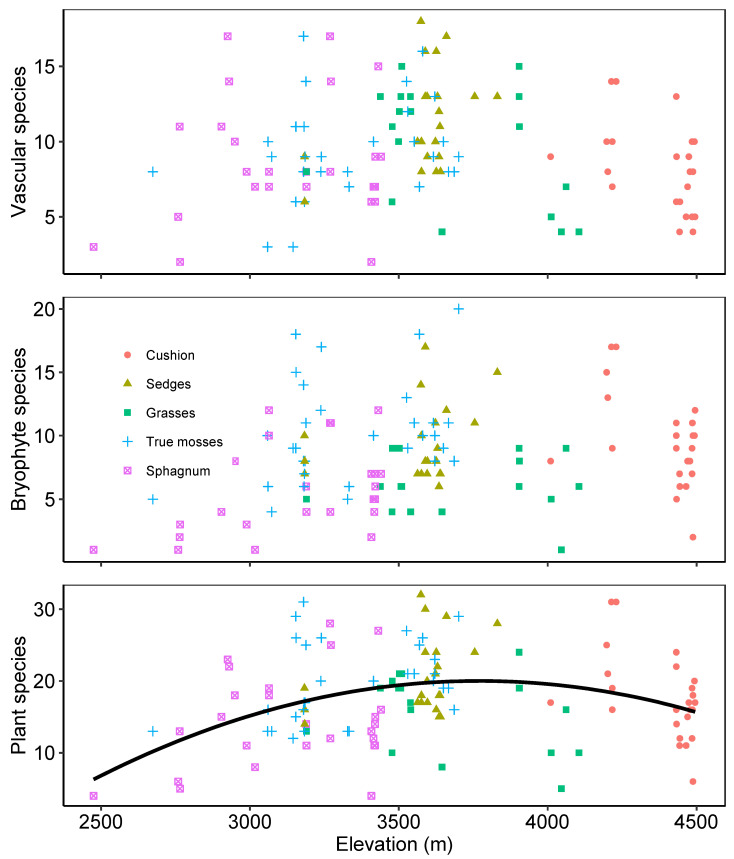
Variation between the number of species at each site and elevation for vascular plants (**top**), bryophytes (**center**) and all plants (**bottom**) differentiated by four different peat-forming vegetation types from 121 peatlands in the northern Andes.

**Figure 2 plants-12-00955-f002:**
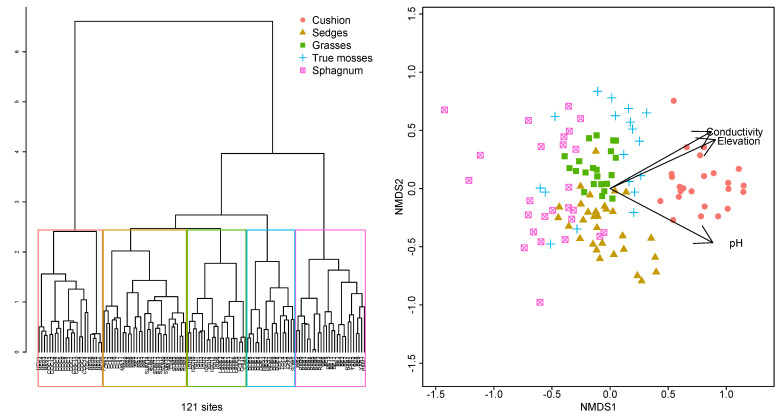
(**Left**) Hierarchical cluster using Ward’s distances on 121 peatlands in the northern Andes indicating the selection of 5 different plant composition groups. (**Right**) NMDS ordination of 121 peatlands in the northern tropical Andes with matching color codes to the 5 cluster groups. Fitted environmental variables in figure had a significant correlation with NMDS ordination after 999 permutations (*p* < 0.01).

**Figure 3 plants-12-00955-f003:**
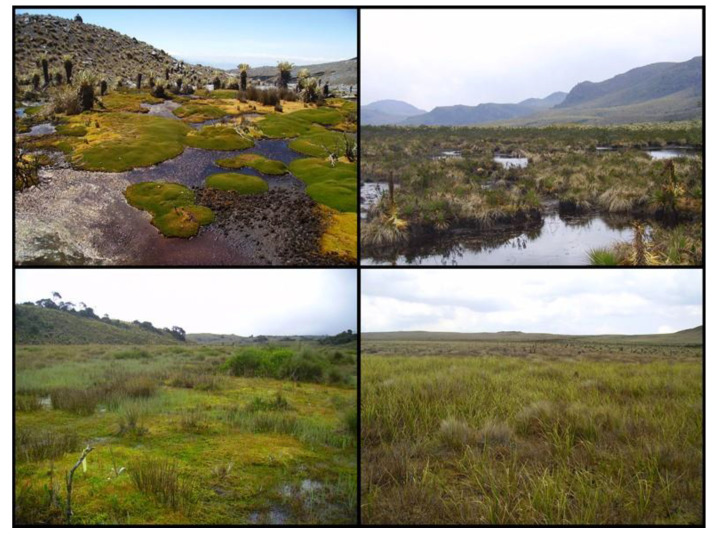
The main peat-forming vegetation types in the northern Andes. (**Top left**): cushion-plant-dominated with *Distichia muscoides* and *Campylopus*; (**top right**): a sedge-dominated site; (**bottom left**): site dominated by *Sphagnum sancto-josephense*; (**bottom right**): a peatland dominated by true mosses and *Calamagrostis effussa*.

**Figure 4 plants-12-00955-f004:**
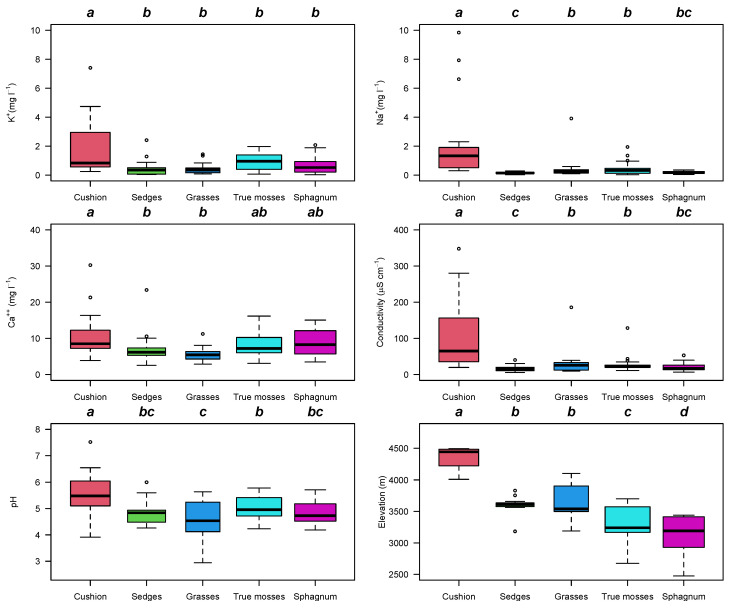
Box and whisker plots for chemistry of pore water variation between the five vegetation types identified in 121 peatlands in the northern Andes. Cation concentrations are presented in mg L^−1^. Water electrical conductivity was corrected for [H^+^] [37]. Tukey’s homogeneous groups are marked with the same letter. The circles represent outlier values outside the 95% interval.

**Figure 5 plants-12-00955-f005:**
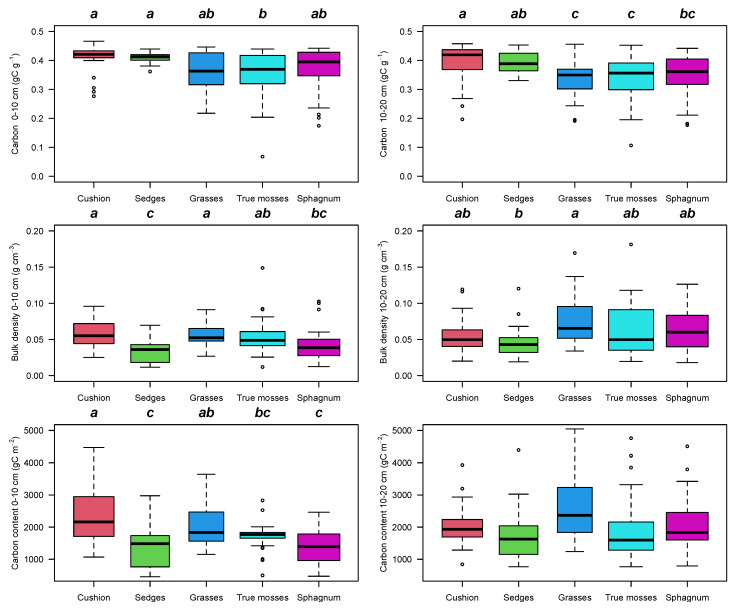
Box and whisker plots for variation in peat carbon concentration, peat bulk density and carbon content for 121 peatlands in the northern Andes for the 0–10 cm and 10–20 cm depth intervals. Tukey’s homogeneous groups are marked with the same letter. The circles represent outlier values outside the 95% interval.

**Figure 6 plants-12-00955-f006:**
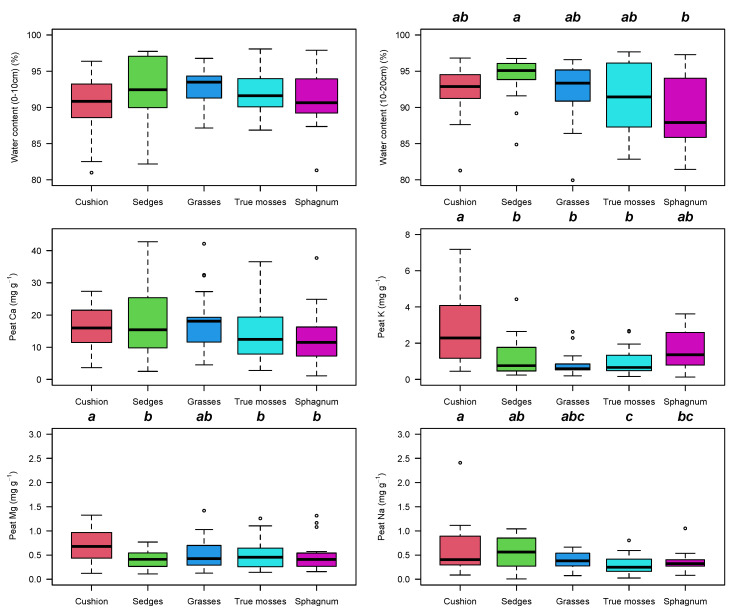
Box and whisker plots for variation in peat water content and peat chemistry for 121 peatlands in the northern Andes. Top row: peat water content as a proportion of the wet peat weight at field capacity subtracting the dry weight. Tukey’s homogeneous groups are marked with the same letter. The circles represent outlier values outside the 95% interval.

**Figure 7 plants-12-00955-f007:**
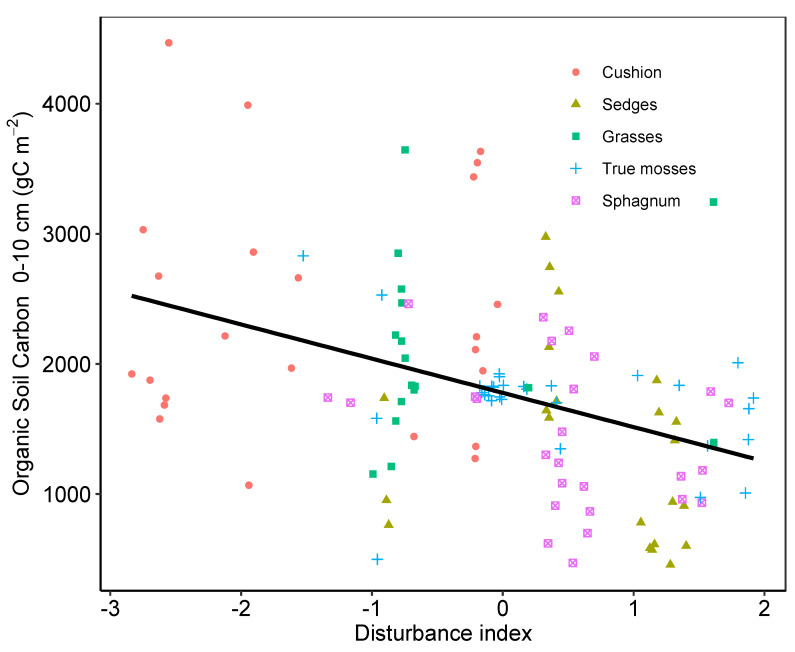
Relationship between carbon content on the 0–10 cm interval and a scaled disturbance observed at each of 121 peatlands in the northern Andes. Higher values of the disturbance index indicate higher evidence of human disturbance in the peatland.

**Figure 8 plants-12-00955-f008:**
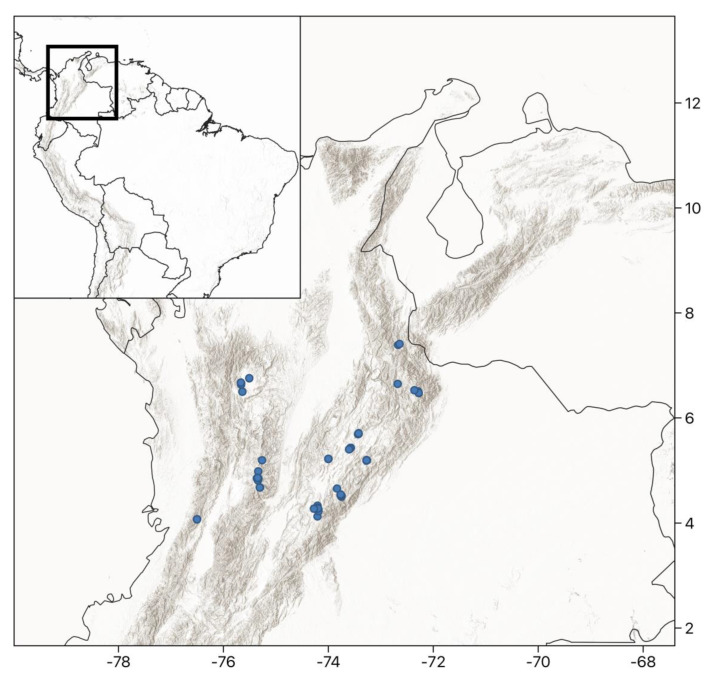
Map of study area in the northern Andes including a shaded digital elevation model (darker areas are higher). Sampled peatlands (*n* = 121) are marked with circles. A number of sites are relatively close to each other, and on this, the points overlap.

**Table 1 plants-12-00955-t001:** Mean percent cover for species with the higher abundance from 121 northern Andean peatlands divided in five main peat-forming vegetation types. Cover values with an asterisk indicate a significant species indicator value for the respective vegetation type using the Dufrene–Legendre method. The number of peatlands assigned to the different vegetation types is indicated by *n*.

Species	Cushion *n* = 24	Sedges *n* = 22	Sphagnum *n* = 26	True mosses *n* = 31	Grasses *n* = 18
*Ageratina tinifolia*	0	0	2 *	1.5	0
*Arcytophyllum muticum*	0	4.3	6	4.4 *	0
*Astrophyllum nigrescens*	3.9 *	0	4	2	1
*Blechnum loxense*	0	2	2.8 *	1.3	0
*Breutelia chrysea*	0	0	0	0 *	0
*Calamagrostis effussa*	2.1	5.4	6.1	3.7	6.4 *
*Campylopus cuspidatus*	3.4	4.4 *	4	4	2.9
*Carex pygmaea*	4.3	5.4	2.8	1.7	11.5 *
*Disterigma alaternoides*	0	1	0	3 *	1
*Distichia muscoides*	18.8 *	0	0	0	6
*Espeletia grandiflora*	0	4.1 *	3	1	2.5
*Espeletia occidentalis*	0	0	0	0	3.7 *
*Hypnum cupressiforme*	1	2	1.7	3.3 *	0
*Juncus effusus*	0	0	1.9 *	1	0
*Lophozia sp*	3 *	0	0	0	1
*Nertera gradensis*	1	2.3	3.5 *	2.8	3
*Oreobolus cleefii*	0	0	0	0	6.7 *
*Paepalanthus colombianus*	0	2.3	2.7 *	2.3	2.8
*Pernettya prostrata*	3	5.7 *	3.6	2	6.4
*Puya goudutia*	0	2.5 *	1.6	1.7	0
*Rhacoccarpus purpurascens*	4.5	4.5	0	4	8.1 *
*Rhynchospora oreoboloidea*	1.9	3.3	0	4.4	6.6 *
*Riccardia paramorum*	5.7	5.9 *	3.6	3.4	3.9
*Sphagnum magellanicum*	0	9.4 *	8.4	4.6	5.5
*Sphagnum sancto-josephense*	8	8.4	12.7	10.6 *	9.8
*Warnstorfia exannulata*	4.6 *	0	0	1	3
*Werneria pygmaea*	6.4 *	0	0	1	3

## Data Availability

The data presented in this study are available on request from the corresponding author. The species data are not publicly available due to specific restrictions from the herbaria. Plant cover and water chemistry data together with R code is available at: https://github.com/jcbenavides/Peatveg.

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
