# Peer review of "The High-Elevation Peatlands of the Northern Andes, Colombia"

_plants, 2023, doi:10.3390/plants12040955_

Round 1

Reviewer 1 Report

This study represents an extensive survey of peatland vegetation in the northern Andes of Colombia. The main results are descriptions of the different plant species present and their relationship to certain variables (i.e. elevation, climate, water chemistry). I believe the information in this study serve as important baseline data in a rapidly changing part of the world and will be of interest to those studying peatlands and environmental change in alpine environments. My comments are largely minor in nature.

Page 3/54, 3rd para. All species names should be in italics.

Page 16/54. “In this study, highly disturbed peatlands had lower carbon content in the upper 10 cm of soil”.

Any idea why this is so? Is this due to the encroachment of upland species that have less carbon content?

Page 16/54. “Peat soil was sampled from 0 to 10 cm and 10 to 20 cm depths using a 10 cm long 5 cm diameter corer.”

What was the reason for sampling two different depths? Were you trying to address any particular questions? Or simply just looking at any possible changes with depth?

Page 16/54,under Analysis. “(tell us what program you used).”

This needs to be completed.

Fig. 4. Minor point, but could you use the same legend samples (for grasses, true mosses, sedges, etc) here as you do in Fig. 2?

Author Response

Reviewer 1

Comments and Suggestions for Authors

This study represents an extensive survey of peatland vegetation in the northern Andes of Colombia. The main results are descriptions of the different plant species present and their relationship to certain variables (i.e. elevation, climate, water chemistry). I believe the information in this study serve as important baseline data in a rapidly changing part of the world and will be of interest to those studying peatlands and environmental change in alpine environments. My comments are largely minor in nature.

Page 3/54, 3rd para. All species names should be in italics.

R/ Font and italics changed in the table

Page 16/54. “In this study, highly disturbed peatlands had lower carbon content in the upper 10 cm of soil”.

Any idea why this is so? Is this due to the encroachment of upland species that have less carbon content?

R/ We don’t know, it is surprising but it may be related to the type of vegetation that comes after disturbance that brings lower carbon or a loss of organic matter via decomposition. We have added that statement to the document.

Page 16/54. “Peat soil was sampled from 0 to 10 cm and 10 to 20 cm depths using a 10 cm long 5 cm diameter corer.”

What was the reason for sampling two different depths? Were you trying to address any particular questions? Or simply just looking at any possible changes with depth?

R/ The corer was 10 cm long, and is now annotated in the methods

Page 16/54,under Analysis. “(tell us what program you used).”

This needs to be completed.

R/ We have clarified the note about the software used for the analysis and the references to the relevant complementary packages

Fig. 4. Minor point, but could you use the same legend samples (for grasses, true mosses, sedges, etc) here as you do in Fig. 2?

R/ figs 2 and 4 share the same legend, Fig 3 has been updated to match

Reviewer 2 Report

Plants_Andes

Title: The high elevation peatlands of the northern Andes, Columbia.

Author: J. C. Benavides

General Comments

The paper examines plant communities in several peatland sites of the northern Andes Mountains. The idea is okay, but the paper is a bit difficult to read. Sorry to be harsh, but the paper reads like an early draft.

I have a few concerns. One is that the site-selection section is a bit too terse. The description of the experimental design needs more detail. You need to provide enough detail for a reviewer to repeat the study. 

Also, I am not convinced that some of the statistics were done correctly; see below.

Thirdly, some of the conclusions are overstated. Perhaps the most egregious is the conclusion that hydrology controls plant community composition. This might be correct, and sorry to be harsh, but the hydrology data presented are thin, at best. See below. 

Keep in mind that is it is essentially impossible to determine cause and effect from a single survey with correlation analysis. The best you can do is speculate. 

The paper has an odd organization. Section 2.4 in the Results is titled ‘Figures, Tables, Schemes.’ Why? 

Lastly, my copy of the paper did not have line numbers. This makes it difficult to covey where editorial changes in needed. I’ll do my best to covey, but it is the author’s problem to follow, i.e., in the future put line numbers on the manuscript. 

Specific Comments

1)    Title. Delete ‘the.’ Also consider saying ‘peatlands of the northern Andes Mountains, Columbia.’ ‘Andes’ is a bit too colloquial for the title. 

2)    If the journal specifies Methods at the end, okay. If not move the Methods to the second section. It is difficult to evaluate Results without knowing how those results were obtained. 

3)    Please provide more information about ‘site selection’ in the Methods. Show there was no bias in site selection. Since you are trying to relate vegetation to environmental conditions, it is important that site selection had some element of randomness. 

4)    For example, in section 4.2 we learn that each vegetation type occurred within each peatland. How was this possible? 

5)    I am no statistician, but the analyses of peat and water chemistry seem simplistic; specifically, the use of a one-way ANOVA. The reason is that you are testing many variables that are not independent of each other. Thus, the likelihood of a Type 1 error exists. Either you need to adjust the alpha with correction, such as a Bonferroni, or consider using a more complicated repeated measures model; each chemical element is a y-value in the model.

6)    The Results section needs reorganization. The problem is that aspects of the peat soil and water chemistry are in section 2.1, before the findings are presented in sections 2.2 and 2.3. Consider a shorter section 2.1 that is just species. The correlations could go in a new section 2. 4. 

7)    The description of disturbance in section 2.3 belongs in the Methods section.

8)    Consider deleting the term ‘vegetation’ from the manuscript. Specifically, ‘vegetation plant communities’ is an awkward term. I realize that mosses and grasses are different vegetation types. But simply noting the differences in plant community composition among sites is good enough. 

9)    The ‘vegetation’ section in the Discussion is an example. This could easily be labeled ‘community composition.’

10) Some of the Discussion seems overstated. For example, I agree that hydrology might influence plant community composition. But the data here are just a one-time sample for water chemistry and water content. A full assessment of hydrology is much more complicated. In other words, does water table level vary seasonally? Annually? Year to year? What is the source water in each site? Rather Discuss the correlations and speculate about hydrology. 

11) Consider a table in the text with some information on common and rare species. This would help to understand groupings. 

12) The discussion of ‘bulk density’ is okay but be cautious. Your measure is actually ‘carbon density’, with units of g C/cm3, not bulk density, which has units of g/cm3. Either use carbon density or bulk (mass) density.

Technical Comments

1)    Abstract: start with ‘peatlands in the Andes Mountains, etc.’ You can use ‘Andean’ after that

2)    Abstract and elsewhere: use a hyphen for ‘peat-forming vegetation.’

3)    Abstract: use a comma after ‘124 bryophytes, etc.’

4)    Abstract: use ‘Sphagnum-dominated.’

5)    Abstract: consider saying, ‘variation in peatland vegetation is mostly driven be elevation and climate, then by water, etc.’

6)    Introduction, first paragraph, line 2: add ‘Mountains’ after ‘Andes.’

7)    Introduction, third paragraph, line 1: consider saying, ‘peatlands are a type of wetland in which, etc.’ 

8)    Introduction, third paragraph, line 3: say ‘mostly anoxic, except at the surface.’

9)    Introduction, third paragraph, line 3: delete the sentence, ‘peat is produced, etc.’ Not all the litter ‘is resistant to decomposition.’ Litter does not decompose because microbial decomposers have slow activity.

10) Introduction, third paragraph, line 16: change ‘strong influence’ to ‘that appear to be influenced by, etc.’

11) Section 4.2, line 1: use a comma after ‘peatland,’

12) Section 4.2, line 14: the sentence, ‘elevation was recorded, etc.’ seems out-of-place.

13) Section 4.2: describe the criteria you used for ‘operational taxonomic unit.’ This is not arbitrary. 

14) Section 4.3, line 2: yes, please ‘tell us what program you used!’

15) Section 2.1, line 2: switch the values, say ‘29% of the species (N = 77) were found at, etc.’

16) Section 2.1, line 7: where are the findings for ‘disturbance.’ This seems like a qualitative measure. 

17) Figure 1: are there really 121 circles on the map? I count about 30. Something is wrong.

18) Figure 2: finish the y-axis, i.e., add ‘(numbers of species).’

19) Figure 4: are we supposed to read the code names on the axis tips? Delete.

20) Figure 5: delete ‘variation’ from the legend. Are there ‘four’ plant communities, or five?

21) Figure 6 and 7 could be combined. 

Author Response

Title: The high elevation peatlands of the northern Andes, Columbia.

Author: J. C. Benavides

General Comments

The paper examines plant communities in several peatland sites of the northern Andes Mountains. The idea is okay, but the paper is a bit difficult to read. Sorry to be harsh, but the paper reads like an early draft.

I have a few concerns. One is that the site-selection section is a bit too terse. The description of the experimental design needs more detail. You need to provide enough detail for a reviewer to repeat the study. 

 R/ we have expanded the description of site selection, understanding that this is the first time peatlands in Colombia are surveyed and local people or the scientific community doesn’t know where we can or can’t find peatlands and the national wetland maps do not include them as a category. An additional filter was that the research permit requested granted the access to the site.

Also, I am not convinced that some of the statistics were done correctly; see below.

 R/ We have see ypur comments below and responded accordingly

Thirdly, some of the conclusions are overstated. Perhaps the most egregious is the conclusion that hydrology controls plant community composition. This might be correct, and sorry to be harsh, but the hydrology data presented are thin, at best. See below. 

  R/ You are correct in that the hydrology data is thin and we tried to use our expirenece and local climatic data to make the inferences. However, this is a geographical study and not long or shor term hydrological data is available from any Andean peatland in the country. The closes we have is Esteban Suarez’s data from Ecuador but their blanket cushion peatlands are not found in Colombia making any transfer of information impossible.

Keep in mind that is it is essentially impossible to determine cause and effect from a single survey with correlation analysis. The best you can do is speculate. 

R/ you are correct and adjusted the language in the paper

The paper has an odd organization. Section 2.4 in the Results is titled ‘Figures, Tables, Schemes.’ Why? 

R/ it is the format of the journal, but we have updated that subtitle

Lastly, my copy of the paper did not have line numbers. This makes it difficult to covey where editorial changes in needed. I’ll do my best to covey, but it is the author’s problem to follow, i.e., in the future put line numbers on the manuscript. 

 R/ again this is the format of the journal and we agree that line numbers are the way to go, we have added them to this version

Specific Comments

1)    Title. Delete ‘the.’ Also consider saying ‘peatlands of the northern Andes Mountains, Columbia.’ ‘Andes’ is a bit too colloquial for the title. 

 R/ we have updated the title. However, we consider that since this is just the first survey from an introductory field study a coloquiall title better describe the work. Just for clarification, the country name is Colombia not Columbia (https://itscolombianotcolumbia.com/)

2)    If the journal specifies Methods at the end, okay. If not move the Methods to the second section. It is difficult to evaluate Results without knowing how those results were obtained. 

 R/ it is the journal organization

3)    Please provide more information about ‘site selection’ in the Methods. Show there was no bias in site selection. Since you are trying to relate vegetation to environmental conditions, it is important that site selection had some element of randomness. 

  R/ done

4)    For example, in section 4.2 we learn that each vegetation type occurred within each peatland. How was this possible? 

 R/ we have removed the first line that was somehow confusing

5)    I am no statistician, but the analyses of peat and water chemistry seem simplistic; specifically, the use of a one-way ANOVA. The reason is that you are testing many variables that are not independent of each other. Thus, the likelihood of a Type 1 error exists. Either you need to adjust the alpha with correction, such as a Bonferroni, or consider using a more complicated repeated measures model; each chemical element is a y-value in the model.

 R/ There are no more factors in the study and the NMDS and cluster analysis shows that the main gradient observed was the vegetation and was mostly driven by elevation and water chemistry. Type error I correction is over the same variable being tested repeatedly. Each variable was tested only once in our analysis of variance.

6)    The Results section needs reorganization. The problem is that aspects of the peat soil and water chemistry are in section 2.1, before the findings are presented in sections 2.2 and 2.3. Consider a shorter section 2.1 that is just species. The correlations could go in a new section 2. 4. 

 R/ we have reorganized the results according to your suggestions, except for the new subsection with the correlations. It will make a section with only one paragraph with no results that can be highlighted during the discussion.

7)    The description of disturbance in section 2.3 belongs in the Methods section.

  R/ we have moved the paragraph according to your suggestion

8)    Consider deleting the term ‘vegetation’ from the manuscript. Specifically, ‘vegetation plant communities’ is an awkward term. I realize that mosses and grasses are different vegetation types. But simply noting the differences in plant community composition among sites is good enough. 

 R/ we have had this discussion internally and we have changed the specific section where we wrote “vegetation plant communities”

9)    The ‘vegetation’ section in the Discussion is an example. This could easily be labeled ‘community composition.’

 R/ we have had this discussion internally and we consider that given the superficial nature of this first survey keeping the work under the framework of vegetation types keep us from trying to define specific communities. We consider that the richness of this work is just to point out the high plant diversity from such a limited data set.

10) Some of the Discussion seems overstated. For example, I agree that hydrology might influence plant community composition. But the data here are just a one-time sample for water chemistry and water content. A full assessment of hydrology is much more complicated. In other words, does water table level vary seasonally? Annually? Year to year? What is the source water in each site? Rather Discuss the correlations and speculate about hydrology. 

 R/ although the data is limited, water chemistry, that is a component of the hydrology fo the sites is clear on that differentiation.  

11) Consider a table in the text with some information on common and rare species. This would help to understand groupings. 

R/ in the supplementary data we have updated the species table including abundance and relevant indicator species

12) The discussion of ‘bulk density’ is okay but be cautious. Your measure is actually ‘carbon density’, with units of g C/cm3, not bulk density, which has units of g/cm3. Either use carbon density or bulk (mass) density.

 R/ we are providing soil bulk density, from which organic carbon is a proportion. That is why we used the product of bulk density and the proportion of organic carbon for the calculation of carbon mass for each depth.

Technical Comments

1)    Abstract: start with ‘peatlands in the Andes Mountains, etc.’ You can use ‘Andean’ after that

R/  Done. However, we have kept the northern Andes name when we refer to conditions not found in the central and southern Andes. I.e. historical use of the peatlands-wetlands

2)    Abstract and elsewhere: use a hyphen for ‘peat-forming vegetation.’

 R/  Done.

3)    Abstract: use a comma after ‘124 bryophytes, etc.’

  R/  Done.

4)    Abstract: use ‘Sphagnum-dominated.’

  R/  Done.

5)    Abstract: consider saying, ‘variation in peatland vegetation is mostly driven be elevation and climate, then by water, etc.’

   R/  Done.

6)    Introduction, first paragraph, line 2: add ‘Mountains’ after ‘Andes.’

    R/  done. “In the northern tropical Andes, mountain peatlands. “

7)    Introduction, third paragraph, line 1: consider saying, ‘peatlands are a type of wetland in which, etc.’ 

   R/  We consider that is better the way it is

8)    Introduction, third paragraph, line 3: say ‘mostly anoxic, except at the surface.’

 R/ done

9)    Introduction, third paragraph, line 3: delete the sentence, ‘peat is produced, etc.’ Not all the litter ‘is resistant to decomposition.’ Litter does not decompose because microbial decomposers have slow activity.

  R/ done, and we have expanded the comment

10) Introduction, third paragraph, line 16: change ‘strong influence’ to ‘that appear to be influenced by, etc.’

  R/ done

11) Section 4.2, line 1: use a comma after ‘peatland,’

  R/ done

12) Section 4.2, line 14: the sentence, ‘elevation was recorded, etc.’ seems out-of-place.

   R/ We have updated the sentence

13) Section 4.2: describe the criteria you used for ‘operational taxonomic unit.’ This is not arbitrary. 

    R/ We have updated the sentence

14) Section 4.3, line 2: yes, please ‘tell us what program you used!’

     R/ We have updated the sentence

15) Section 2.1, line 2: switch the values, say ‘29% of the species (N = 77) were found at, etc.’

      R/ We consider that the change is not necessary

16) Section 2.1, line 7: where are the findings for ‘disturbance.’ This seems like a qualitative measure. 

       R/ Disturbance is referred in a single sentence at the end of the results. The disturbance assessment was a combination of over 30 different disturbance indicators that were both quantitative and qualitative but we consider that a lengthy description of the disturbance assessment was unnecessary given the focus of the paper.

17) Figure 1: are there really 121 circles on the map? I count about 30. Something is wrong.

 R/ they are superimpose given the scale of the country. A site is less than a 100 ha and the map covers around 80.000.0000 ha. We have updated the caption.

18) Figure 2: finish the y-axis, i.e., add ‘(numbers of species).’

 R/ Done

19) Figure 4: are we supposed to read the code names on the axis tips? Delete.

  R/ They are for reference, we prefer to keep them, since in a online version you can zoom to the labels.

20) Figure 5: delete ‘variation’ from the legend. Are there ‘four’ plant communities, or five?

R/ Done, we differentiated five vegetation types

21) Figure 6 and 7 could be combined. 

R/ We tried that but the figures just don’t match

Reviewer 3 Report

Thank you for the opportunity to review the manuscript. It contains many exciting data that may be important and useful for understanding the diversity and development of the flora of Andean peatlands. 

The study results will interest a wide range of wetland specialists and, as such, potentially merit publication. I want to make a few comments that could contribute to improving this piece of work. 

In the last sentences of the Introduction, the authors write about three research questions and mention only two of them (unless the third one is hidden in point no. 2. 

The authors write that their "research is focused on understanding how northern Andean peatland vegetation types vary along the combined gradients of water chemistry and climate." Unfortunately, it is difficult to find in the paper an analysis of environmental gradients (except for elevation), given as the rate of change of a given scalar physical quantity with respect to the position coordinates.

2 Results

I would remove or clarify the ambiguous phrase "intermediate elevations."

Please use the abbreviation m.a.s.l. instead of masl

The results of the cluster analysis are pretty obvious. The cutoff of clusters (Fig. 4) seems artificial and subordinated to the need to obtain five plant groups. 

In the heading of Table 1, after the period ending the sentence "...Dufrene-Legendre method", n should be written with a capital letter.

How was the "pore water" extracted? Such information needs to be included in the methodology.

The sentence - "The mineral concentration in peat was higher in cushion plants than in the other vegetation types for K, Mg, and Na (K F4,108 = 7.2, p < 0.001, F4,111 = 3.7, p = 0.007, and F4,110 = 4.74, p = 0.001) (Fig. 7) - is unclear   

Fig. 2 - regression lines are not provided for Bryophytes and Vascular species.

Fig. 6 - the designation of homogeneous statistically different groups is unclear - if the authors distinguish groups, for example: "a" and "c", and there is no group "b", then where does the group "ab" or "bc" come from?

4. Materials and methods - "(tell us what program you used)".

Author Response

Thank you for the opportunity to review the manuscript. It contains many exciting data that may be important and useful for understanding the diversity and development of the flora of Andean peatlands. 

The study results will interest a wide range of wetland specialists and, as such, potentially merit publication. I want to make a few comments that could contribute to improving this piece of work. 

In the last sentences of the Introduction, the authors write about three research questions and mention only two of them (unless the third one is hidden in point no. 2. 

    R/ Thank you, we have updated the sentence

The authors write that their "research is focused on understanding how northern Andean peatland vegetation types vary along the combined gradients of water chemistry and climate." Unfortunately, it is difficult to find in the paper an analysis of environmental gradients (except for elevation), given as the rate of change of a given scalar physical quantity with respect to the position coordinates.

    R/ We don’t understand this comment, perhaps some clarification will help us to see how can we improve our paper. Although, we did run an spatial autocorrelation analysis (Moran’s I) that didn’t bring significant insights besides that there is an autocorrelation patter at a distance that is equivalent to the distance between cordilleras.

2 Results

I would remove or clarify the ambiguous phrase "intermediate elevations."

    R/ Thank you, we have updated the sentence

Please use the abbreviation m.a.s.l. instead of masl

R/ done

The results of the cluster analysis are pretty obvious. The cutoff of clusters (Fig. 4) seems artificial and subordinated to the need to obtain five plant groups. 

R/ In the results we described a previous analysis of k means were the five groups division has the highest information value. See methods and results. (line 15 page 3)

In the heading of Table 1, after the period ending the sentence "...Dufrene-Legendre method", n should be written with a capital letter.

R/ done

How was the "pore water" extracted? Such information needs to be included in the methodology.

R/ done

The sentence - "The mineral concentration in peat was higher in cushion plants than in the other vegetation types for K, Mg, and Na (K F4,108 = 7.2, p < 0.001, F4,111 = 3.7, p = 0.007, and F4,110 = 4.74, p = 0.001) (Fig. 7) - is unclear   

    R/ We have updated the sentence

Fig. 2 - regression lines are not provided for Bryophytes and Vascular species.

    R/ Because they are not significant and it is described in the text

Fig. 6 - the designation of homogeneous statistically different groups is unclear - if the authors distinguish groups, for example: "a" and "c", and there is no group "b", then where does the group "ab" or "bc" come from?

    R/ A standard practice with Tukey post hoc test is to label the gropus to which the treatments belong (or their means). This a common practice in statistics and  the combination of ab or bc incidates that the particular treatment or level of that treatment can’t be separated from the groups a and b. We consider that this doesn’t require an additional description and can be found on the referenced literature in aprticular the package Agricolae in R (de Mendiburu, 2021)

  1. Materials and methods - "(tell us what program you used)".

R/ Done

Round 2

Reviewer 3 Report

The manuscript can be accepted for publication